# DC-STAMP activates the PI3K/AKT/mTOR signaling pathway to regulate PANoptosis in acute myeloid leukemia

Qian Liang[1], Biao Li[2], Yue Li[3], Longhui Ma[3], Li Dong[1], Ning Ding[3], Wei Zhang[3], Haoran Wang[3], Junying Liu[4]*

1 Department of Hematology, Zhoukou Central Hospital, Zhoukou Medical Science Research Center, Zhoukou, China, 2 Department of Cerebral Interventional Therapy, Zhoukou Central Hospital, Zhoukou Medical Science Research Center, Zhoukou, China, 3 Teaching Management Section, Zhoukou Central Hospital, Zhoukou Medical Science Research Center, Zhoukou, China, 4 Department of Gastroenterology, Zhoukou Central Hospital, Zhoukou Medical Science Research Center, Zhoukou, China

* kjk1796@163.com

## Abstract

### Background

PANoptosis is a newly defined form of programmed cell death that integrates features of apoptosis, pyroptosis and necroptosis, playing a critical role in immune regulation and tumor biology. Clinically, Acute Myeloid Leukemia (AML) patients with high DC-STAMP expression exhibited notably poorer cytogenetic risk profiles and shorter overall survival. Gene set enrichment analysis of primary AML samples from public databases revealed significant enrichment of the mTORC1 signaling pathway, a core signaling axis regulating the apoptotic process, in AML samples with high DC-STAMP expression.

### Methods

DC-STAMP knockdown and overexpression models were established in the AML cell line THP-1 using small interfering RNA (siRNA) and lentiviral plasmids, respectively. Western blotting and RT-PCR were used to assess changes in PI3K/AKT/mTOR pathway activity in response to altered DC-STAMP expression. Flow cytometry and other cellular phenotypic assays were employed to evaluate the impact of DC-STAMP on PANoptosis in AML cells. Finally, PI3K inhibitors were introduced to assess the functional reversal of DC-STAMP–driven malignant phenotypes through downstream PI3K pathway inhibition.

### Results

High DC-STAMP expression in AML activated the PI3K/AKT/mTOR signaling pathway and suppressed the PANoptosis process, thereby enhancing leukemic cell

**Data availability statement:** All data generated or analyzed in this study are included in the present manuscript.

**Funding:** This study was supported by Science and Technology Development Plan Projects of Henan Province in 2024 / Henan Province Science and Technology Research Projects (No. 242102310152), Henan Province Medical Science and Technology Research Project (Joint Construction Program) in 2024 (No. LHGJ20240999) and Key Project of Medical Science and Technology Research Program of Zhoukou Central Hospital in 2023 (No. 20230102). The funders had no role in study design, data collection and analysis, decision to publish, or preparation of the manuscript.

**Competing interests:** The authors have declared that no competing interests exist.

survival and chemoresistance. In contrast, genetic silencing of DC-STAMP or pharmacological inhibition of downstream PI3K restored normal apoptotic processes and significantly attenuated the malignant phenotypes driven by mTOR hyperactivation.

## Conclusions

Activation of DC-STAMP is an essential mechanism that suppresses PANoptosis and promotes chemoresistance in AML cells. Targeting the downstream PI3K/mTOR signaling pathway may offer a promising therapeutic strategy for this high-risk AML subtype.

---

## Introduction

Acute Myeloid Leukemia (AML) is a highly aggressive malignancy of the hematopoietic system, defined by abnormal proliferation of immature myeloid cells and accompanied by high relapse rates and chemotherapeutic resistance. Current standard therapies—including chemotherapy, targeted therapy, and hematopoietic stem cell transplantation—can induce short-term complete remission in some patients. However, the overall cure rate remains unsatisfactory, especially for those with high-risk mutations or malignant fusions, whose five-year survival rate is below 30% [1–4]. Therefore, identifying novel therapeutic targets remains a critical priority. In recent years, a regulatory cell death mode involving features of apoptosis, pyroptosis, and necroptosis known as PANoptosis has been highlighted for its potential role in tumor immune regulation and drug resistance [5]. Studies suggest that dysregulated PANoptosis may play a key role in AML relapse and chemoresistance [5–7]. The mTOR signaling pathway, a central regulator of PANoptosis, has attracted significant research interest [8]. Therefore, elucidating the molecular mechanisms governing PANoptosis in AML could improve therapeutic efficacy and prognosis. Nevertheless, research on the PANoptosis regulatory network in AML remains scarce.

Dendritic Cell-Specific Transmembrane Protein (DC-STAMP) is a risk factor related with poor prognosis in AML, with significantly activated in AML patients compared to other cancers. Within AML cohorts, DC-STAMP overexpression correlates with advanced age, high-risk molecular markers, and reduced survival rates [9]. Gene Set Enrichment Analysis (GSEA) of DC-STAMP-high AML samples [10] revealed marked activation of the mTORC1 signaling pathway, suggesting that DC-STAMP activation may upregulate mTORC1 to promote malignant proliferation and suppress PANoptosis, thereby accelerating AML progression and chemoresistance. DC-STAMP, a key regulator of osteoclast fusion, also plays a vital role in dendritic cell immune homeostasis. Studies show that DC-STAMP deficiency blocks osteoclast and macrophage fusion, leading to increased bone density, while its expression in dendritic cells regulates cytokine secretion and T-cell activation to maintain self-antigen tolerance [9,11–13]. However, DC-STAMP's hydrophobic properties caused by multiple transmembrane domains (e.g., α-helix and β-barrel



conformations) and its dependence on normal physiological functions make direct targeting challenging: DC-STAMP inhibition may disrupt bone metabolism and immune homeostasis [14,15], while its membrane-embedded conformation hinders small-molecule drug binding [16,17]. These limitations underscore the need for alternative downstream targets. In this study, we established DC-STAMP-overexpressing and knockdown models in the THP-1 AML cell line using lentiviral plasmids and siRNA respectively [18]. By combining these models with the PI3K inhibitor LY294002 [19], we demonstrate for the first time that DC-STAMP suppresses PANoptosis and promotes AML cell survival and drug resistance through activation of the PI3K/AKT/mTOR signaling pathway. Further studies confirmed that inhibiting PI3K, a key node in this pathway, effectively reverses DC-STAMP-driven malignant proliferation and apoptosis resistance. Our findings not only fill the research gap regarding the role of DC-STAMP in AML pathogenesis and PANoptosis regulation but also identify a potential therapeutic target for DC-STAMP-high AML patients, offering new strategies to overcome chemoresistance and improve prognosis.

## Materials and methods

### Gene set enrichment analysis (GSEA)

All the sequencing data in this study was publicly available and were downloaded from the TCGA-AML dataset (https://portal.gdc.cancer.gov/). Data were not pre-filtered based on any underlying genetic abnormalities. Public AML samples were then ranked by DC-STAMP expression from highest to lowest, and those falling within the top 50% of expression were designated as "DC-STAMP high-expression AML" samples for subsequent GSEA.

### Cell culture

THP-1 acute myeloid leukemia suspension cells (ATCC) were cultured in complete RPMI-1640 medium (Gibco, USA) supplemented with 10% fetal bovine serum (FBS; Gibco, USA) and 1% penicillin–streptomycin (Gibco, USA) at 37°C in a humidified incubator with 5% $CO_2$ and 95% relative humidity. When the cell density reached $2 \times 10^6$/mL, cells were passaged at a ratio of 1:3.

### Cell transfection

The human DC-STAMP CDS (NM_030788.4: nt 50–1462) was retrieved from NCBI (Broad Institute Portal). Three siRNAs were designed as follows (sense/antisense 5′→3′):

siRNA-1: AGACUUAGGAAGAUAUCAGUG/ CUGAUAUCUUCCUAAGUCUUU;

siRNA-2: AAAGACUUAGGAAGAUAUCAG/ GAUAUCUUCCUAAGUCUUUGG;

siRNA-3: AAAAUGUAUGGAAAAGCUCUU/ GAGCUUUUCCAUACAUUUUCC;

si-scrambled: UUCUCCGAACGUGUCACGUTT/ ACGUGACACGUUCGGAGAATT.

The full-length DC-STAMP CDS was PCR-amplified and cloned into the pcDNA3.1(+) vector (Thermo Fisher Scientific, USA) to generate pcDNA3.1-OE-DC-STAMP, synthesized by Sangon Biotech (Shanghai) (Thermo Fisher Scientific, US) (Fig 1). For the knockdown model, a scrambled siRNA (si-scrambled) was used as the negative control. For the overexpression model, the empty pcDNA3.1(+) vector (Empty) was transfected and used as the corresponding control. When the experiment includes both overexpression (OE) and knockdown (KD) groups, we used cells treated only with the transfection reagent (Veh contrl) in culture medium as the control group. Cells at a density of $2 \times 10^6$/mL were passaged once 24h before transfection to ensure optimal health. According to the matching protocol, diluted Lipofectamine 3000 and DNA/P3000 mixtures (Thermo Fisher Scientific, USA) were combined 1:1, incubated 10–15 min at RT, then added to 6-well plates for 48 h.

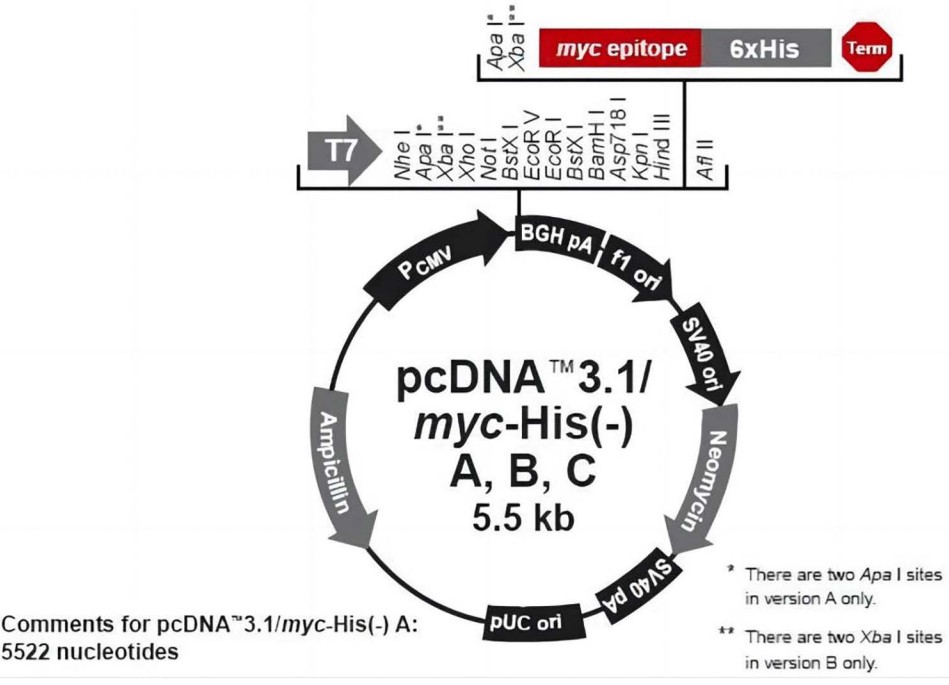

**Fig 1. The plasmid map of pcDNA3.1.**

## CCK-8 assay

THP-1 cells were devided into following groups based on different treatments: 1.Control (untreated), 2.DC-STAMP knockdown 3.DC-STAMP overexpression 4.LY294002 (Selleck Chemicals, USA) (10µM) 5.DC-STAMP overexpression+LY294002 (10µM). Cells were rinsed with PBS and plated at a density of $2 \times 10^3$cells per well in 96-well plates, then cultured for 48h at 37°C in a 5% $CO_2$incubator. All experimental conditions were set up in triplicate to ensure reproducibility. After the incubation, 10 µL of CCK-8 reagent (Yeasen, China) was added to each well, and the cells were further incubated for 2 hours to allow sufficient reaction between the CCK-8 reagent and the cells. Finally, the absorbance at 450 nm was measured using a microplate reader (Thermo Fisher Scientific, USA) to quantify cell viability. The absorbance value of the blank well containing only water was used as the baseline and subtracted from all readings. The readings of all experimental groups were then normalized to control group.

## Immunofluorescence

Cells ($1 \times 10^6$ per well) were seeded in 6-well plates, washed once with PBS (Gibco, USA), and fixed in 1 mL 4% paraformaldehyde (50 rpm, 20 min, RT), then washed three times with PBS. After permeabilization in 0.25% Triton X-100 (Sigma-Aldrich, USA) (1 mL, 20 min, RT) and a single PBS wash, samples were blocked in 1 mL goat serum for 30 min at RT. Primary antibodies (p-AKT, p-mTOR, Caspase-3, Cell Signaling Technology, USA) were diluted in blocking buffer and incubated overnight at 4°C, followed by three 5-min TBST washes. Fluorophore-conjugated secondary antibody (1:200) was applied for 1 h at RT in the dark, washed thrice with TBST, then nuclei were counterstained with DAPI (Beyotime, China) (5 min, RT), washed once with PBS, and mounted in Fluoroshield® with DAPI. Stained cells were imaged by fluorescence microscopy and quantified using ImageJ.

## qRT-PCR

Total RNA was extracted from cells using the FastPure Total RNA Kit with gDNA Filer and RNA Columns (Vazyme, China), following sequential buffer and ethanol washes at 12,000 g for 30 s per step. Reverse transcription was carried out in a 20 μL reaction using the HiScript III 1st Strand cDNA Synthesis Kit with gDNA wiper (Vazyme, China): gDNA was removed at 42°C for 2 min, after which the RT Mix and enzyme were added and incubated at 37°C for 15 min, followed by 85°C for 5 s. The resulting cDNA was stored at –20°C. Quantitative PCR was performed in 20 μL reactions using 2×Taq Pro Universal SYBR qPCR Master Mix (Vazyme, China) on a Bio-Rad CFX96 instrument (Bio-Rad, USA). The cycling conditions were: 95°C for 30 s; 40 cycles of 95°C for 10 s and 60°C for 10 s; followed by a melt curve analysis at 95°C for 15 s, 65°C for 60 s, and 95°C for 15 s. Relative gene expression was calculated using the $2^{-\Delta\Delta Ct}$ method. Primer sequences are listed in Table 1.

## Western blot

Cells ($1 \times 10^6$ per well) were lysed in RIPA buffer containing PMSF (Beyotime, China) and centrifuged at 12,000 g for 5 min at 4°C. Protein concentrations were determined using the Pierce BCA Protein Assay Kit (Thermo Fisher Scientific, USA) in a microplate format (sample: working reagent, 1:20). Samples were mixed with 5×loading buffer (Beyotime, China), boiled for 10 min, and resolved on 12% SDS-PAGE gels with 5% stacking gels (Bio-Rad, USA). Proteins were then transferred to PVDF membranes (Millipore, USA), which were blocked with 5% milk in TBST (Sigma-Aldrich, USA) for 30 min at room temperature. Membranes were incubated with primary antibodies (Cell Signaling Technology, USA) overnight at 4°C, washed, and then incubated with HRP-conjugated secondary antibodies (Cell Signaling Technology, USA) for 2 h at room temperature. Protein bands were visualized using ECL reagents (Millipore, USA) and imaged on a JP-K6000 system (Jiapeng, China). Densitometric analysis was performed using ImageJ (NIH, USA), with target bands normalized to GAPDH.

## Flow cytometry

Cells ($1 \times 10^6$) were collected, washed twice in ice-cold PBS, and resuspended in 100 μL of 1×Annexin V Binding Buffer (BD Biosciences, USA). FITC-Annexin V (BD Biosciences, USA) (5 μL) and propidium iodide (BD Biosciences, USA) (5 μL) were then added, the suspension was gently mixed, and samples were incubated in the dark at room temperature for 15 minutes. After staining, 400 μL of 1×Binding Buffer was added, and cells were analyzed within 1 hour on a BD FACS-Canto II cytometer. Data acquisition used BD FACSDiva software, with analysis in FlowJo v10. Early apoptotic cells were defined as Annexin $V^+/PI^-$, and late apoptotic/necrotic cells as Annexin $V^+/PI^+$.

**Table 1. The sequence of primers.**

| Primers | Sequence (5'→3') | |
|---|---|---|
| DC-STAMP | Forward | ACTTATCCATCTCTGCATCTGG |
| | Reverse | AAGGATAGAGGACAACAGTCC |
| PI3K | Forward | CAACAATTACGCGCTCAGACAT |
| | Reverse | GCGCGTAATTGTTGTTTTTGACTG |
| AKT | Forward | TACGCGCTTGAGCTCATCC |
| | Reverse | ACACGATACCGGCAAAGAAG |
| mTOR | Forward | ATAAACAACAATTACGCGCTCCAG |
| | Reverse | CAATTCTGCAGATGAGCACATC |
| GAPDH | Forward | AAAATCAAGTGGGGCGATGC |
| | Reverse | GCGCGTAATTGTTGTTTACACC |

### Venetoclax resistance assay

DC-STAMP-overexpressing THP-1 cells in the logarithmic growth phase were seeded in six-well plates at $5 \times 10^6$ cells per well. Cells were treated with a gradient of different concentrations of venetoclax (Selleck Chemicals, USA) in the culture medium and incubated for 48 h. After treatment, cells were harvested, and apoptosis levels were assessed by flow cytometry following the standard apoptosis detection protocol.

### Statistical analysis

Data are expressed as mean ±SD. Statistical analysis was performed using GraphPad Prism 9.0. Comparisons between two groups used Student's t-test. Significance thresholds were set at *$P < 0.05$, **$P < 0.01$, and ***$P < 0.001$.

## Results

### Establishment and validation of DC-STAMP overexpression and knockdown THP-1 cell models

Dendritic Cell-Specific Transmembrane Protein (DC-STAMP) is an independent risk factor that leads to poor prognosis in acute myeloid leukemia (AML). We first downloaded and analyzed DC-STAMP expression profiles from the TCGA database across all genetic backgrounds of AML patients [20]. Samples were grouped into High-DC-STAMP and Low-DC-STAMP groups based on the median expression level. GSEA results of the high-DC-STAMP group identified 28 significantly enriched pathways, including the mTORC1 signaling pathway (Normalized Enrichment Score = 2.15, FDR < 0.05), which regulates cellular apoptosis and is closely associated with leukemia development (Fig 2A). mTORC1, as a hub for cellular energy and nutrient sensing, regulates apoptotic decisions by integrating growth factors, amino acids, and energy status. Its Rheb-dependent activation on the cytoplasmic and lysosomal membranes enables the mTORC1 complex to recruit and phosphorylate downstream effectors, thereby controlling cell survival-death balance [21]. Given the essential role of mTORC1 in apoptosis regulation, we established stable DC-STAMP knockdown (KD) and overexpression (OE) THP-1 cell models using siRNA and lentiviral plasmids to systematically investigate the DC-STAMP-mTORC1 regulatory axis. qRT-PCR showed significantly reduced DC-STAMP mRNA levels in the KD group compared to the negative control group (P < 0.001), while OE group exhibited markedly elevated expression (P < 0.001) (Fig 2B). Western blot analysis demonstrated that DC-STAMP protein expression was markedly reduced to 22.5% of NC levels in the KD group (P < 0.01), whereas the OE group exhibited a 4.58-fold increase relative to empty control (P < 0.001) (Fig 2C). These results validate the reliability of the cell models for subsequent experiments. In addition, our previous work demonstrated that high DC-STAMP expression in AML is associated with poor prognosis [9]. Using the established overexpression cell model, we further confirmed that THP-1 cells with DC-STAMP overexpression exhibited marked resistance to venetoclax compared with the control group (Fig 2D). This further underscores the significance of investigating its pathogenic mechanisms and therapeutic potential.

### The aberrant expression of DC-STAMP regulates apoptosis in AML cells

After successfully constructing and validating DC-STAMP knockdown and overexpression THP-1 cell models, we performed phenotypic assays including CCK-8 and flow cytometry to verify the GSEA results and investigate DC-STAMP's effects on AML cell proliferation and survival. THP-1 cells from each group were cultured under standard conditions for 72h. CCK-8 assays showed that cell viability was significantly reduced in the KD group compared to the control group (P < 0.01), while markedly increased in the OE group (Fig 3A), and this trend became progressively more pronounced over the 72-hour period. Apoptosis was assessed by Annexin V-FITC/PI staining and flow cytometry. DC-STAMP knockdown (KD) significantly increased THP-1 cell apoptosis at 24 h, 48 h, and 72 h compared with the control (Veh) (24 h: ~17% vs. ~14%; 48 h: ~29% vs. ~14%; 72 h: ~28% vs. ~13%; all P < 0.001), with the most pronounced elevation observed at 48 h and 72 h. In contrast, DC-STAMP overexpression (OE) markedly decreased apoptosis at the same time points (24



**Fig 2. Establishment and validation of DC-STAMP overexpression and knockdown THP-1 cell models. (A)** GSEA revealed significant enrichment of the mTORC1 pathway in DC-STAMP-high AML samples. (NES = 2.591, FDR q < 0.05, adjusted p = 0.008). **(B)** qRT-PCR validation of relative DC-STAMP mRNA expression levels in knockdown and overexpression THP-1 cells, normalized to the control group. **(C)** Western blot analysis of

DC-STAMP protein expression in KD and OE groups (β-actin as loading control). Right panel: Quantitative densitometry of protein bands. **(D)** Flow cytometry was used to assess apoptosis in control and DC-STAMP-overexpressing THP-1 cells after 48 hours of treatment with different concentrations of vincristine. Each result represents the mean of three independent biological replicates. Error bars represent mean±SD). *P<0.05, **P<0.01, ***P<0.001 (two-tailed t-test).

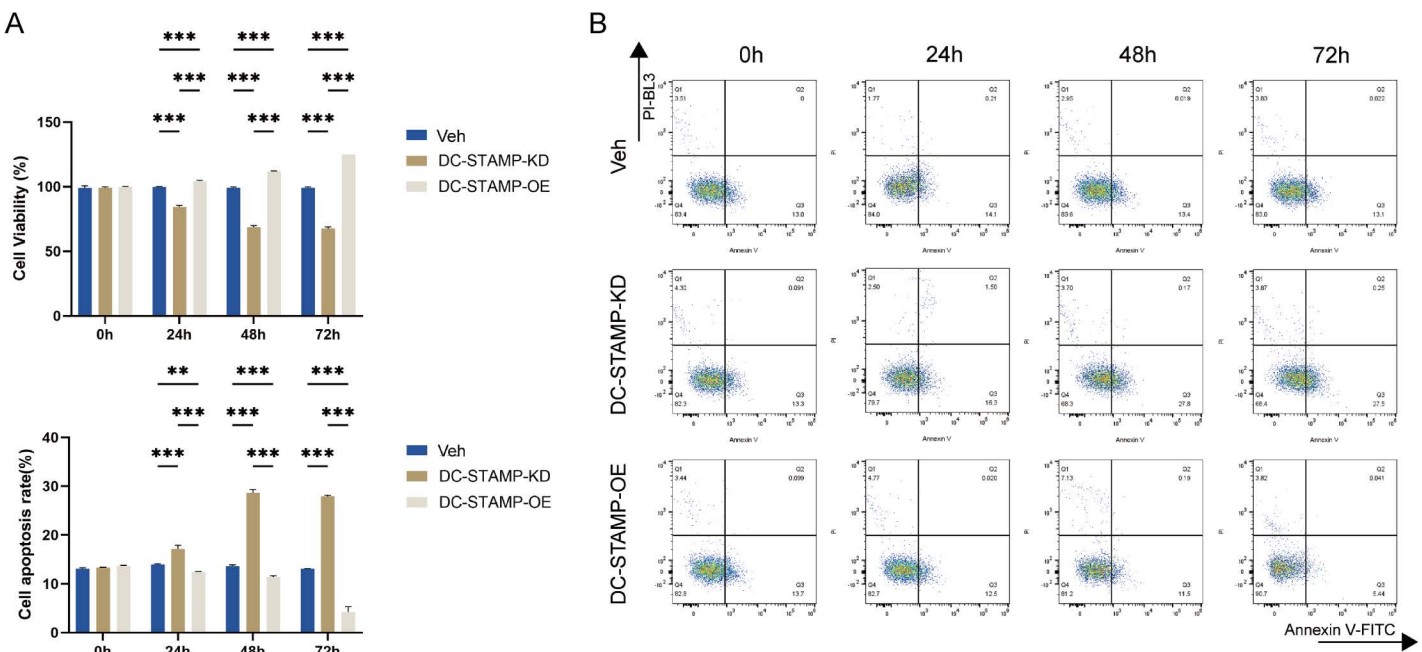

**Fig 3. The aberrant expression of DC-STAMP regulates apoptosis in AML cells. (A)** Cell viability of THP-1 cells in each group after 0h/24h/48h/72h of DC-STAMP knockdown or overexpression, measured using the CCK-8 assay. Data are normalized to the NC group (error bars represent mean±SD). **(B)** Apoptosis levels assessed by Annexin V-FITC/PI double staining and flow cytometry after after 0h, 24h, 48h,72h of DC-STAMP knockdown or overexpression. Bar graph shows the total apoptosis rate (early+late apoptosis) in each group. Each result represents the mean of three independent biological replicates. Error bars represent mean±SD). *P<0.05, **P<0.01, ***P<0.001 (two-tailed t-test).

h:~12% vs.~14%; 48 h:~11% vs.~14%; 72 h:~4% vs.~13%; all P<0.001), indicating that DC-STAMP overexpression suppresses apoptosis. Overall, apoptosis exhibited a time-dependent increase in the KD group, whereas it progressively declined in the OE group (Fig 3B). Together, these results demonstrate that DC-STAMP promotes AML cell proliferation and suppresses apoptosis, while its knockdown reduces cell viability and enhances apoptotic cell death.

## DC-STAMP exerts an upstream regulatory function on the mTOR signaling pathway

After validating the impact of DC-STAMP on AML cell survival, we further investigated whether DC-STAMP exerts this function by acting as an upstream regulator of the mTOR signaling pathway. We first performed immunofluorescence to detect the intracellular localization and activation levels of key signaling factors p-AKT and p-mTOR in THP-1 cell models. Results showed that compared to the control group, the fluorescence intensity of p-AKT (green) and p-mTOR (red) was significantly reduced in the DC-STAMP knockdown group (P<0.01), indicating that DC-STAMP knockdown effectively inhibits PI3K/AKT/mTOR pathway activation. Conversely, the DC-STAMP overexpression group exhibited significantly enhanced fluorescence intensity of these phosphorylated proteins (P<0.001), demonstrating DC-STAMP's role in promoting pathway activation (Fig 4A).

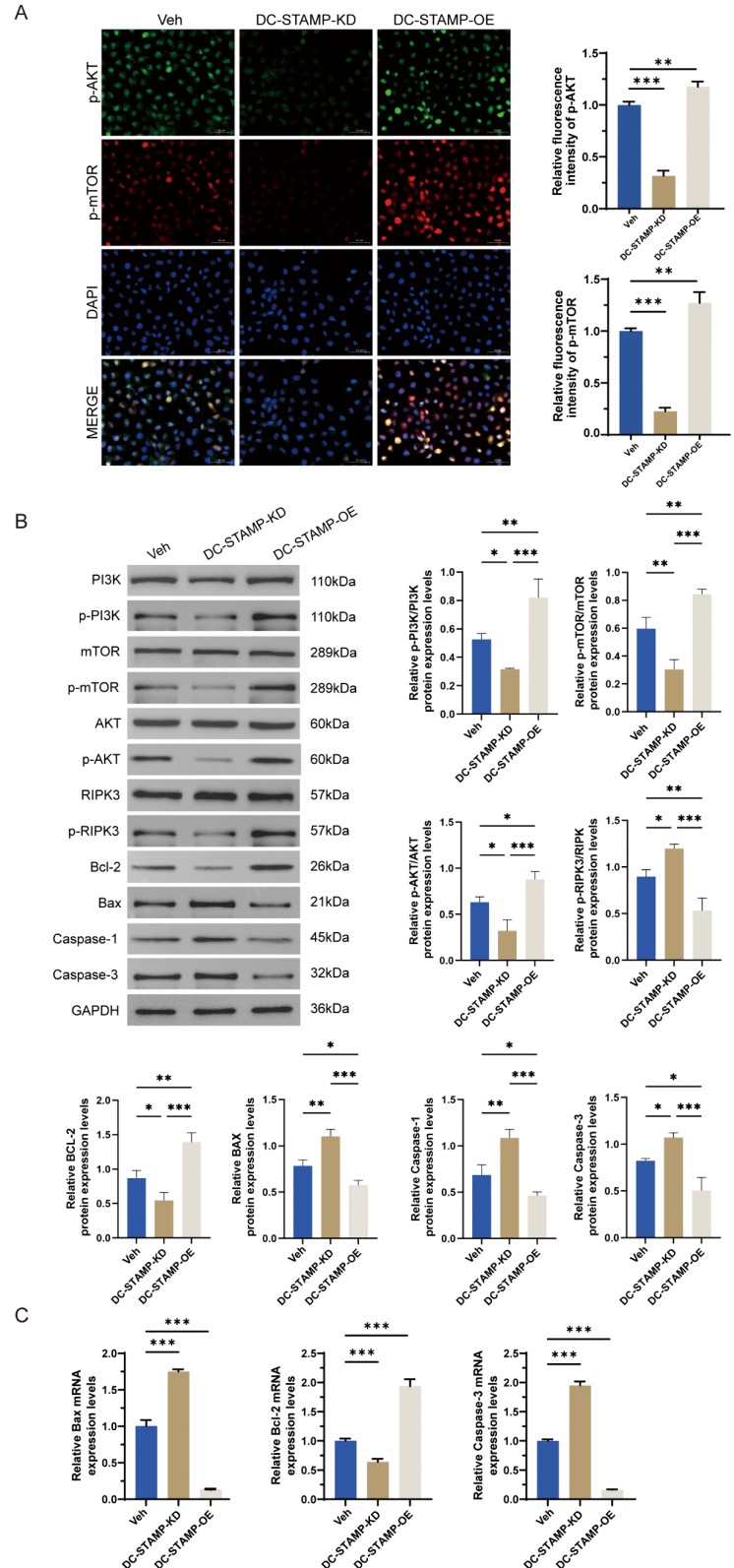

**Fig 4. DC-STAMP exerts an upstream regulatory function on the mTOR signaling pathway. (A)** Immunofluorescence analysis of PI3K/AKT/ mTOR pathway activation in THP-1 cells. Subcellular localization of phosphorylated AKT (p-AKT, Ser473, green) and phosphorylated mTOR (p-mTOR,

Ser2448, red), with nuclear counterstaining by DAPI (blue). Scale bar: 20 µm. The lower panel shows quantitative analysis of fluorescence intensity (n = 3; error bars represent mean ± SD). **(B)** Expression levels of pro-apoptotic genes (Bax and Caspase-3), the anti-apoptotic gene Bcl-2, and pan-apoptotic/necroptotic/pyroptotic markers (RIPK3 and Caspase-1) were normalized to GAPDH (n = 3; error bars represent mean ± SD). **(C)** Western blot analysis of PI3K/AKT/mTOR signaling and apoptosis-associated proteins. Protein levels of PI3K, p-PI3K (Tyr458), mTOR, p-mTOR (Ser2448), AKT, p-AKT (Ser473), and apoptosis regulators Bcl-2, Bax, and Cleaved Caspase-3 were detected. β-actin served as the internal control. The right panel shows densitometric quantification of protein band intensity. Each result represents the mean of three independent biological replicates. Error bars represent mean ± SD). *P < 0.05, **P < 0.01, ***P < 0.001 (two-tailed t-test).

To confirm these findings at the protein level, we conducted quantitative Western blot analysis of phosphorylation levels at key nodes of the PI3K/AKT/mTOR pathway. Compared to the Vehicle control (Veh) group, the KD group showed significantly reduced expression of p-PI3K, p-AKT, and p-mTOR (P < 0.01), as well as decreased levels of the pan-apoptotic pathway marker p-RIPK3 (P < 0.01); in contrast, the OE group exhibited significantly increased expression of these phosphorylated proteins (P < 0.001) (Fig 4B). These results are highly consistent with the immunofluorescence data, further confirming that DC-STAMP overexpression activates the PI3K/AKT/mTOR pathway, while DC-STAMP inhibition suppresses its activity. Additionally, qRT-PCR analysis of apoptosis-related genes revealed that the KD group had down-regulated Bcl-2 mRNA levels (P < 0.001) and upregulated levels of pro-apoptotic genes Bax and Caspase-3 (P < 0.001). Conversely, the OE group showed upregulated Bcl-2 and downregulated Bax and Caspase-3 (P < 0.001) (Fig 4C). These data demonstrate that DC-STAMP coordinates the expression of multiple apoptosis-related genes, leading to imbalanced cell proliferation and promoting malignant expansion of AML cells. These findings demonstrate that DC-STAMP acts as a key regulator of AML cell survival, sustaining malignant proliferation mainly through activation of the PI3K/AKT/mTOR pathway and suppression of apoptosis. Targeting DC-STAMP or its downstream signaling may therefore provide therapeutic potential in AML.

### Inhibition of the downstream PI3K rescues DC-STAMP activation-induced malignant proliferation in AML

As a prototypical transmembrane protein, the transmembrane domains (α-helix or β-barrel conformations) of DC-STAMP exhibit high hydrophobicity, with their core regions naturally embedded within the lipid bilayer [14,15]. This structural feature prevents the ligand-binding interface from effectively interacting with conventional small-molecule inhibitors in aqueous environments. Such characteristics significantly limit the binding capacity and targeting efficacy of small-molecule rugs against DC-STAMP [22]. Given these limitations, we investigated an alternative therapeutic strategy: indirectly inhibiting DC-STAMP's oncogenic function by targeting downstream signaling nodes. Notably, PI3K—a key upstream kinase of the mTOR pathway—has shown promising targeted inhibitory effects and clinical potential in treating various malignancies [23,24]. To validate this strategy, we treated DC-STAMP-overexpressing THP-1 cells with 10 µM of the PI3K-specific inhibitor LY294002 for different time points and systematically assessed its ability to rescue DC-STAMP-driven malignant phenotypes. Our results showed that in DC-STAMP-overexpressing THP-1 cells, LY294002 treatment significantly reduced the abnormal increase in cell viability (Fig 5A) and the aberrant activation of the PI3K/AKT/mTOR pathway induced by DC-STAMP overexpression, and partially restored the expression of pro-apoptotic genes (Bax and Caspase-3) (Fig 5B-C). Furthermore, LY294002 reversed the anti-apoptotic effect of DC-STAMP, restoring apoptosis rates in DC-STAMP-overexpressing cells to the baseline levels observed in wild-type cells (Fig 5D). In summary, as showed in the graphical abstract in Fig 6, our results indicate that elevated DC-STAMP expression regulates PANoptosis in AML cells by activating the mTORC1 signaling pathway, leading to apoptosis resistance and drug tolerance; targeting the downstream PI3K node effectively counteracts the pro-leukemogenic effects of DC-STAMP overexpression, providing a potential therapeutic alternative for DC-STAMP–activated AML when direct targeting is technically challenging.



**Fig 5. Inhibition of the downstream PI3K rescues DC-STAMP activation-induced malignant proliferation in AML. (A)** Cell viability of THP-1 cells in each group and different time points, measured using the CCK-8 assay. Data are normalized to the negative control group (NC) (n = 3; error bars represent mean ± SD). **(B)** Immunofluorescence analysis of PI3K/AKT/mTOR pathway activation. Subcellular localization of phosphorylated AKT (p-AKT, Ser473, green) and phosphorylated mTOR (p-mTOR, Ser2448, red), with nuclei stained by DAPI (blue). Scale bar: 20 μm. The lower panel shows quantitative analysis of fluorescence intensity (n = 3; error bars represent mean ± SD). **(C)** Western blot analysis of PI3K/AKT/mTOR signaling and apoptosis-related proteins. Expression levels of PI3K, p-PI3K (Tyr458), mTOR, p-mTOR (Ser2448), AKT, p-AKT (Ser473), and apoptosis regulators Bcl-2, Bax, and Cleaved Caspase-3 were detected. β-actin was used as the loading control. The lower panel presents densitometric quantification of protein band intensity (n = 3; error bars represent mean ± SD). **(D)** Apoptosis levels assessed by Annexin V-FITC/PI double staining and flow cytometry in the Vehicle control, DC-STAMP overexpression (DC-STAMP-OE), DC-STAMP-OE + 10 μM LY294002 (DC-STAMP-OE + LY), and Empty control + 10 μM LY294002 (Empty+PI3K-LY) groups. The lower panel shows quantitative results of total apoptosis rate (early + late apoptosis). Each result represents the mean of three independent biological replicates. Error bars represent mean ± SD). *P < 0.05, **P < 0.01, ***P < 0.001 (two-tailed t-test).

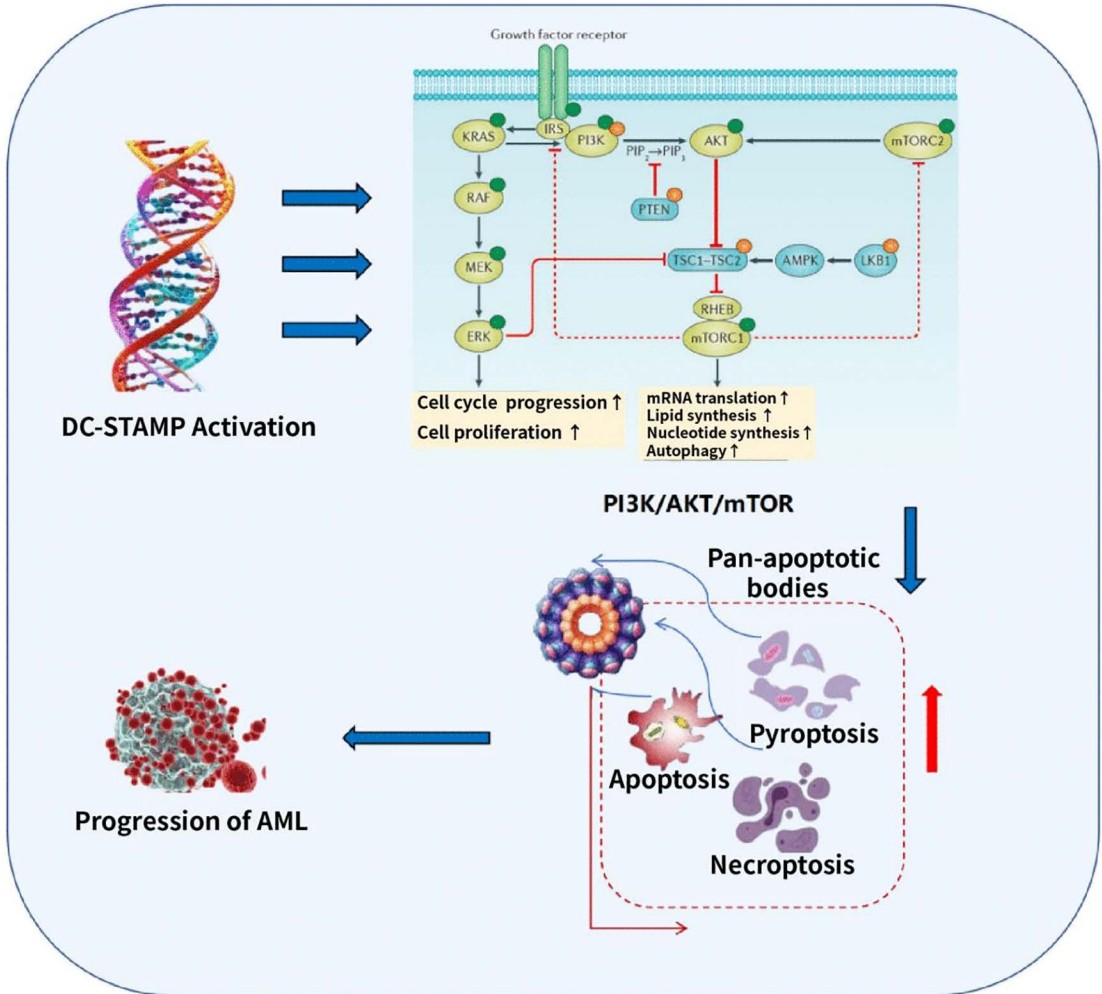

**Fig 6. High DC-STAMP expression in AML suppresses PANoptosis by upregulating the PI3K/AKT/mTOR signaling pathway, thereby accelerating leukemia progression.**

## Discussion

Although most acute myeloid leukemia (AML) patients can achieve rapid complete remission after standard chemotherapy, the high relapse rates and chemoresistance driven by high-risk factors are still challenging. Beyond classical mitochondrial-dependent apoptosis, necroptosis, pyroptosis, and the newly defined PANoptosis have been implicated in leukemia drug resistance and immune evasion [25–28]. As a coordinated cell death modality integrating features of apoptosis, pyroptosis, and necroptosis, PANoptosis is regulated by the PANoptosome complex and profoundly impacts cancer progression and therapeutic responses [29]. However, research on PANoptosis and its regulatory factors in AML remains critically underexplored. In previous study, we confirmed in TCGA-AML cohort that high DC-STAMP expression was significantly associated with shortened overall survival across all AML genetic subtypes, including patients classified as ELN2022 low-risk. Within both MLL-rearranged and NPM1-mutated subgroups, DC-STAMP overexpression independently predicted poorer prognosis, establishing it as a universal adverse biomarker irrespective of underlying genetics [9]. And here by using DC-STAMP-overexpressing/knockdown AML cell models, we found DC-STAMP knockdown significantly

reduced PI3K/AKT/mTOR pathway activation (decreased p-AKT and p-mTOR) and upregulated pro-apoptotic genes such as BAX and Caspase-3, while downregulating anti-apoptotic Bcl-2 [30]. Conversely, DC-STAMP overexpression induced hyperactivation of PI3K/AKT/mTOR signaling and reversed apoptosis-related gene expression patterns, confirming DC-STAMP's critical role in suppressing PANoptosis and driving chemoresistance in AML. Also, we explored a therapeutic strategy targeting PI3K with inhibitors to reverse the malignant phenotype driven by DC-STAMP activation, thereby identifying an alternative target for the untargetable DC-STAMP. Preclinical data also indicate that combining PI3K inhibitors (e.g., copanlisib) with BCR-ABL inhibitors (imatinib) or chemotherapeutic agents (doxorubicin) enhances tumor-cell apoptosis and overcomes drug resistance [31,32]. Together, these findings provide both empirical and theoretical support for incorporating PI3K inhibitors into precision-medicine regimens for DC-STAMP–high AML.

The PI3K/AKT/mTOR pathway plays an essential role in the leukemogenesis by regulating cell proliferation, survival, and metabolism [33]. Two independent researches have demonstrated that hyperactivation of PI3K/AKT/mTOR contributes to treatment failure in high-risk AML but did not identify upstream triggers [34,35]. Our work fills this gap by placing DC-STAMP upstream of PI3K/AKT/mTOR, revealing a membrane-bound sensor capable of rapid signal-complex assembly and amplification of pro-survival cascades. Morever, regarding that few studies have addressed the activation mechanism of DC-STAMP, we hypothesize that DC-STAMP may sense extracellular ligands or adhesion signals via its ectodomain, subsequently recruiting the PI3K p85 regulatory subunit at its intracellular tail to activate the p110 catalytic subunit, thereby triggering AKT/mTORC1 phosphorylation cascades. This predicted model aligns with DC-STAMP's capacity for rapid membrane-bound signal complex assembly and explains its spatial amplification mechanism for cell fate determination [36,37]. However, further studies should be done to validate this hypothesis through co-immunoprecipitation (Co-IP) or cryo-electron microscopy. In addition, this study is limited to in vitro cell models and retrospective TCGA data, lacking in vivo validation, which is also the main task that urgently remains to be undertaken.

## Conclusions

Through multi-omics and mechanistic analyses, we show that DC-STAMP suppresses PANoptosis in AML cells via activation of the PI3K/AKT/mTOR pathway, promoting cell survival and chemoresistance. Given the difficulty of directly targeting DC-STAMP, we identified the downstream PI3K axis as an alternative therapeutic target to induce PANoptosis. These findings highlight the translational and clinical potential of this strategy.

## Supporting information

**S1 File. Raw images.**
(PDF)

## Author contributions

**Conceptualization:** Qian Liang, Junying Liu.

**Data curation:** Qian Liang, Longhui Ma, Ning Ding, Wei Zhang, Junying Liu.

**Formal analysis:** Qian Liang, Biao Li, Yue Li, Ning Ding, Junying Liu.

**Funding acquisition:** Junying Liu.

**Investigation:** Haoran Wang.

**Methodology:** Qian Liang, Junying Liu.

**Project administration:** Junying Liu.

**Resources:** Junying Liu.

**Software:** Biao Li, Yue Li, Longhui Ma, Li Dong.



**Supervision:** Biao Li, Yue Li, Longhui Ma.

**Validation:** Qian Liang.

**Visualization:** Qian Liang.

**Writing – original draft:** Qian Liang, Biao Li, Junying Liu.

**Writing – review & editing:** Qian Liang, Biao Li, Yue Li, Longhui Ma, Li Dong, Ning Ding, Wei Zhang, Haoran Wang, Junying Liu.

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
