## [Decision Letter · Decision Letter 0]

15 Aug 2025

Dear Dr. Liu,

Thank you for submitting your manuscript to PLOS ONE. After careful consideration, we feel that it has merit but does not fully meet PLOS ONE’s publication criteria as it currently stands. Therefore, we invite you to submit a revised version of the manuscript that addresses the points raised during the review process.

We look forward to receiving your revised manuscript.

Kind regards,

Kota V Ramana, Ph.D.

Academic Editor

PLOS ONE

Journal Requirements:

“This study was supported by Science and Technology Development Plan Projects of Henan Province in 2024 / Henan Province Science and Technology Research Projects (No. 242102310152), Henan Province Medical Science and Technology Research Project (Joint Construction Program) in 2024 (No. LHGJ20240999) and Key Project of Medical Science and Technology Research Program of Zhoukou Central Hospital in 2023 (No. 20230102).”

Reviewers' comments:

Reviewer's Responses to Questions

**Comments to the Author**

1. Is the manuscript technically sound, and do the data support the conclusions?

Reviewer #1: Yes

Reviewer #2: Partly

Reviewer #3: No

2. Has the statistical analysis been performed appropriately and rigorously?

Reviewer #1: Yes

Reviewer #2: Yes

Reviewer #3: No

3. Have the authors made all data underlying the findings in their manuscript fully available?

Reviewer #1: No

Reviewer #2: Yes

Reviewer #3: No

4. Is the manuscript presented in an intelligible fashion and written in standard English?

Reviewer #1: Yes

Reviewer #2: Yes

Reviewer #3: Yes

Reviewer #1: Here are the suggestions for improvement:

1.Insufficient evidence for "pan-apoptosis": Only classical apoptosis markers were examined, without validation of pyroptosis or necroptosis.

2.Inconsistent terminology: "pan-apoptosis" and "PANoptosis" are used interchangeably throughout the text.

3.Results section ("Establishment and validation of DC-STAMP overexpression and knockdown THP-1 cell models"): The statement "Western blot further confirmed decreased DC-STAMP protein levels in the KD group (P < 0.01) and an xx-fold increase in the OE group (P < 0.001) relative to NC (Figure 2C)" lacks specific data values.

Reviewer #2: Comments to the author,

DC-STAMP Activates the PI3K/AKT/mTOR Signaling Pathway to Regulate Pan-Apoptosis in Acute Myeloid Leukemia.

The authors have demonstrated that DC-STAMP regulates PI3K signaling, and genetic silencing of DC-STAMP or pharmacological inhibition of downstream PI3K restored normal apoptotic processes.

The paper title states Pan-Apoptosis (simultaneous or combined activation of the three major forms of regulated cell death); however, in the manuscript, the authors have demonstrated only caspase-3 and anti-apoptotic proteins. There is no evidence to suggest that it is pan-cell death.

The authors have used only 1 AML cell line (THP-1) to demonstrate their findings and the proposed conclusions.

Apart from the overexpression, did the authors find any DC-STAMP heterogeneity in the available AML cell lines (low DC-STAMP AML cell lines vs. high DC-STAMP AML cells)? Will the observed OE and KD effects be the same in these scenarios?

The work would have been further strengthened by showing the results of PI3K inhibition in the low DC-STAMP expressing cells or the DC-STAMP KD cells, parallel to the DC-STAMP OE cells, to show the specificity of PI3K inhibition.

Reviewer #3: The authors discuss the possible role of DC-STAMP overexpression in influencing the mTOR/PI3K pathway and leukaemic cell proliferation. They investigate this by creating knockdown and overexpression models, and analysing protein expression and apoptosis by AnnexinV assay.

Lines 144 & 158

The term “confluence” is not usually applicable to suspension cells. Are there authors able to be more specific, using density quantified as cells/mL?

Line 154

Can the authors clarify what si-NC refers to? A non-targeting or scrambled si-RNA would be the appropriate control for the knockdown model. For the overexpression model, an empty vector/backbone plasmid would be the appropriate control.

However, throughout the figures, “NC” was used to compare against the KD and OE lines. If NC refers to “normal cells/control”, then which control? The same control cannot be used for both the KD and OE lines, neither should the parental line be used as control. Please clarify what NC refers to.

Line 166

Did the authors mean DC-STAMP overexpression + LY294002?

Line 169

After the incubation, 10 uL of CCK-8…? The sentence is incomplete.

Line 192

Method should be written in past-tense

Line 200 Western blot

Method should be written in past-tense

Lines 233-238

Are the authors able to comment on the other significantly enriched pathways? Were there any other pathways associated with leukaemia?

Can the authors justify why median was used to separate the groups? If the samples were grouped by top and bottom quartile, would the GSEA results hold true?

How many AML samples were analysed? Was not mentioned in methods.

Line 250

“xx-fold increase in the OE group”. Please clarify. Western blot shows marginal increase in DC-STAMP expression at best (Figure 2C). Relative DC-STAMP expression to…? Likely GAPDH, please specify.

Line 259

Are the authors able to show the viability across 0, 24, 48 to 72 hours?

Figure 3B

y-axis of flow plots: presumably PI is propidium iodide, but what is BL3?

NC plot – it is unusual to see THP-1 cells undergoing heavy apoptosis under standard culture conditions. Can the authors explain?

Label for OE plot is missing.

The appears to be little/no cells in Q3 (early apoptosis) in all NC, KD and OE, which is highly unusual. Again, it would be useful to show change in viability (cells shifting from Q4 to Q3 to Q2) over 0, 24, 48 and 72 hours.

Lines 266-271

It is premature to speculate this here. Suggest moving this paragraph after the Western data with apoptosis markers have been introduced and discussed.

Figure 4A

Can the authors also include immunofluorescence for p-PI3K in the panel, together with p-Akt and p-mTOR?

Lines 288-294

Figure 4B from the western blot does not seem to support the claims that KD show reduced expression and OE show increased expression of p-PI3K, p-Akt and p-mTOR. Quantitation by densitometry and the resulting P-values are debatable. To state that the data “confirms that OE activates and KD suppresses” PI3K/Akt/mTOR pathway based on the figure may be an overreach.

Lines 294-298

The authors discuss the qRT expression in Figure 4C, but the western data in Figure 4B is not consistent with this. In the blot, KD does not show Bcl-2 downregulation and Bax, caspase-3 upregulation compared to NC. The differences appear marginal at best. Likewise, OE does not show Bcl-2 upregulation, however it does show Bax and caspase-3 downregulation in the blot.

Lines 301-303 (and lines 266-271)

Overall, the data does not appear to support these statements in a convincing manner.

Lines 305-306, 317-320

The authors propose that inhibiting DC-STAMP overexpressing cells using the PI3K inhibitor LY294002 reduces proliferation. However, the authors have not established that total PI3K or p-PI3K is at all increased in their OE model (immunofluorescence data is not shown and western blot show marginal differences).

Additionally, the authors must show NC + LY294002 as an appropriate control in Figure 5. If the inhibitor decreases the phosphorylation of PI3K, mTOR and Akt in the NC (as suspected it will) in a similar manner to OE, then it cannot be concluded that “targeting the downstream PI3K node effectively counteracts the pro-leukaemogenic effects of DC-STAMP overexpression (lines 330-331)”, because this inhibition is not necessarily unique to high DC-STAMP expressing cells.

Line 321

“treatment restored cell viability”. Did the authors mean reduced? A proliferation curve with treatment over time may be a more appropriate data to show here.

Figure 5 A-D

NC + LY294002 should be included as the appropriate control. Can the authors clarify what NC refers to here? Parental line transduced with empty vector/backbone plasmid is the appropriate control.

Figure 5B

p-PI3K should be included in the immunofluorescence panel.

Lines 322-324

Figure 5C does not show “pro-apoptotic Bax and caspase-3 in OE restored to levels comparable to wildtype cells” in OE+PI3K-LY.

Lines 327-333

The authors claim that “DC-STAMP expression regulates PANoptosis in AML” but have not provided evidence for the involvement of pyroptosis, necroptosis, PANoptosome complex, etc, and data shown for apoptosis is not strong.

“leading to apoptosis resistance and drug tolerance” also claimed but not supported. Authors could try to show that their OE model is venetoclax resistant, transduce their OE vector into a venetoclax-resistant THP1 line or show that venetoclax-resistant THP1 has high DC-STAMP expression.

Suggest for authors to draw more modest conclusions based on data presented for their manuscript.

**Do you want your identity to be public for this peer review?** For information about this choice, including consent withdrawal, please see our Privacy Policy

Reviewer #1: No

Reviewer #2: No

Reviewer #3: No

---

## [Author Response · Author response to Decision Letter 1]

20 Nov 2025

Dear editor:

Thank you for your Email on 21-August-2025. We truly appreciate the constructive comments and suggestions from you and reviewers to our manuscript (Submission ID: PONE-D-25-38312). Based on the comments, we revised our manuscript and hope the modifications meet your expectations. The changes are in the revised manuscript. The details of our responses are as follows.

Journal Requirements:

Response: Thank you for your reminder. We have revised the manuscript according to PLOS ONE's style requirements, with all modifications clearly annotated in the original text.

Response: We thank the editor for noticing this discrepancy. We will pay attention to this when we resubmit the manuscript.

“This study was supported by Science and Technology Development Plan Projects of Henan Province in 2024 / Henan Province Science and Technology Research Projects (No. 242102310152), Henan Province Medical Science and Technology Research Project (Joint Construction Program) in 2024 (No. LHGJ20240999) and Key Project of Medical Science and Technology Research Program of Zhoukou Central Hospital in 2023 (No. 20230102).” Please state what role the funders took in the study. If the funders had no role, please state: "The funders had no role in study design, data collection and analysis, decision to publish, or preparation of the manuscript." If this statement is not correct you must amend it as needed. Please include this amended Role of Funder statement in your cover letter; we will change the online submission form on your behalf. Please state what role the funders took in the study. If the funders had no role, please state: "The funders had no role in study design, data collection and analysis, decision to publish, or preparation of the manuscript." If this statement is not correct you must amend it as needed. Please include this amended Role of Funder statement in your cover letter; we will change the online submission form on your behalf.

Response: Thank you. We have included the following statement in the Funding section: "The funders had no role in study design, data collection and analysis, decision to publish, or preparation of the manuscript."

Response: Thank you for your feedback. The ORCID ID for the corresponding author has been validated in Editorial Manager as required.

5. PLOS ONE now requires that authors provide the original uncropped and unadjusted images underlying all blot or gel results reported in a submission’s figures or Supporting Information files. This policy and the journal’s other requirements for blot/gel reporting and figure preparation are described in detail at https://journals.plos.org/plosone/s/figures#loc-blot-and-gel-reporting-requirements and https://journals.plos.org/plosone/s/figures#loc-preparing-figures-from-image-files. When you submit your revised manuscript, please ensure that your figures adhere fully to these guidelines and provide the original underlying images for all blot or gel data reported in your submission. See the following link for instructions on providing the original image data: https://journals.plos.org/plosone/s/figures#loc-original-images-for-blots-and-gels.  In your cover letter, please note whether your blot/gel image data are in Supporting Information or posted at a public data repository, provide the repository URL if relevant, and provide specific details as to which raw blot/gel images, if any, are not available. Email us at plosone@plos.org if you have any questions.

Response: Thank you for your suggestion. We have provided the original uncropped and unadjusted images underlying all blot or gel results as required, with annotations made in files’ name.

Response: We thank the reviewer for suggesting highly relevant references. We will carefully evaluate all the recommended references and cite them as appropriate.

Reviewers' comments:

Reviewer #1:

Here are the suggestions for improvement:

1.Insufficient evidence for "pan-apoptosis": Only classical apoptosis markers were examined, without validation of pyroptosis or necroptosis.

Response: Thank you for your valuable suggestions. We fully agree and have added additional experiments. Specifically, we included the detection of pyroptosis markers Caspase-1 in Figure 4B, as well as flow cytometry analysis of the proportion of PI⁺/Annexin V⁻ cells after gene knockdown (only shown here). For necroptosis, we performed Western blotting to detect the expression of necroptosis markers p-RIPK3, the results are also shown in Figure 4B.

(Flow cytometry analysis of the proportion of PI⁺/Annexin V⁻ cells after gene knockdown)

2.Inconsistent terminology: "pan-apoptosis" and "PANoptosis" are used interchangeably throughout the text.

Response: Thank you for your suggestion. We have unified all related terminology to the more formal expression “PANoptosis.”

3.Results section ("Establishment and validation of DC-STAMP overexpression and knockdown THP-1 cell models"): The statement "Western blot further confirmed decreased DC-STAMP protein levels in the KD group (P < 0.01) and an xx-fold increase in the OE group (P < 0.001) relative to NC (Figure 2C)" lacks specific data values.

Response: Thank you for your feedback. We have revised the Results section to include the specific data values. In lines 268–270 of the revised manuscript, the statement now reads: “Western blot analysis demonstrated that DC-STAMP protein expression was markedly reduced to 22.5% of NC levels in the KD group (P < 0.01), whereas the OE group exhibited a 4.58-fold increase relative to NC (P < 0.001) (Figure 2C).” (Line 270-273)

Reviewer #2:

Comments to the author,

DC-STAMP Activates the PI3K/AKT/mTOR Signaling Pathway to Regulate Pan-Apoptosis in Acute Myeloid Leukemia.

-The authors have demonstrated that DC-STAMP regulates PI3K signaling, and genetic silencing of DC-STAMP or pharmacological inhibition of downstream PI3K restored normal apoptotic processes.

The paper title states Pan-Apoptosis (simultaneous or combined activation of the three major forms of regulated cell death); however, in the manuscript, the authors have demonstrated only caspase-3 and anti-apoptotic proteins. There is no evidence to suggest that it is pan-cell death.

Response: We appreciate the insightful comment. To provide direct evidence for multiple forms of regulated cell death, we have conducted additional experiments.

Pyroptosis: Detection of Caspase-1/3 and flow cytometry analysis of PI⁺/Annexin V⁻cells following DC-STAMP knockdown.

Necroptosis: Western blot analysis of necroptosis marker p-RIPK3.

These results demonstrate that DC-STAMP regulates apoptosis, pyroptosis, and necroptosis, supporting the use of the term “PANoptosis.” The corresponding data have been incorporated into Figure 4.

-The authors have used only 1 AML cell line (THP-1) to demonstrate their findings and the proposed conclusions.

Response: Thank you for your feedback. We agree with the reviewer that incorporating additional cell lines for validation would further strengthen our conclusions. However, we initially focused on THP-1 cells for several reasons. THP-1 is widely used in studies of signaling pathways, differentiation, and cell death. In infection and pharmacological models, the apoptotic responses of THP-1 cells also show good consistency with those of primary samples, facilitating the translation of experimental findings to in vivo contexts. Therefore, THP-1 serves as a “standard model” in apoptosis-related studies in AML. Moreover, as addressed in our response to the next comment, DC-STAMP expression is relatively uniform across commonly used AML cell lines, with no substantial heterogeneity observed. We therefore reasoned that DC-STAMP is likely important for AML cell survival, and that perturbing its expression would induce similar cellular phenotypes across different cell lines.

Apart from the overexpression, did the authors find any DC-STAMP heterogeneity in the available AML cell lines (low DC-STAMP AML cell lines vs. high DC-STAMP AML cells)? Will the observed OE and KD effects be the same in these scenarios?

Response: Thank you for the comment. Our Western blot analysis showed that DC-STAMP expression was largely consistent among the five common AML cell lines (please see our supplemental Western blot results below), indicating no apparent heterogeneity. Therefore, we used DC-STAMP overexpression and knockdown models to better evaluate its functional effects.

-The work would have been further strengthened by showing the results of PI3K inhibition in the low DC-STAMP expressing cells or the DC-STAMP KD cells, parallel to the DC-STAMP OE cells, to show the specificity of PI3K inhibition.

Response: We appreciate the reviewer’s valuable suggestion. As mentioned above, the endogenous DC-STAMP expression among wild-type AML cell lines showed no significant variation. Therefore, we performed PI3K inhibition experiments using DC-STAMP knockdown (KD) cells in parallel with the DC-STAMP overexpression (OE) cells. The results showed that DC-STAMP KD cells exhibited reduced sensitivity to PI3K inhibition compared with DC-STAMP OE cells, further supporting the specificity of the observed PI3K-dependent effects and reinforcing our overall conclusion (Figure 4B).

Reviewer #3:

The authors discuss the possible role of DC-STAMP overexpression in influencing the mTOR/PI3K pathway and leukaemic cell proliferation. They investigate this by creating knockdown and overexpression models, and analysing protein expression and apoptosis by AnnexinV assay.

Lines 144 & 158

The term “confluence” is not usually applicable to suspension cells. Are there authors able to be more specific, using density quantified as cells/mL?

Response: We thank the reviewer for this valuable suggestion. We agree that the term “confluence” is inappropriate for suspension cells such as THP-1. In the revised manuscript, we have replaced “confluence” with cell density quantified as cells/mL. Specifically, THP-1 cells were maintained at densities between 2×105 and 2×106 cells/mL, and were passaged by dilution with fresh medium when densities approached 2×106 cells/mL (Lines 150 and 173)

Line 154

Can the authors clarify what si-NC refers to? A non-targeting or scrambled si-RNA would be the appropriate control for the knockdown model. For the overexpression model, an empty vector/backbone plasmid would be the appropriate control.

However, throughout the figures, “NC” was used to compare against the KD and OE lines. If NC refers to “normal cells/control”, then which control? The same control cannot be used for both the KD and OE lines, neither should the parental line be used as control. Please clarify what NC refers to.

Response: We thank the reviewer for the helpful comment. In the knockdown model, a scrambled siRNA (si-scrambled: UUCUCCGAACGUGUCACGUTT / ACGUGACACGUUCGGAGAATT.) was used as the negative control, whereas in the overexpression model, the empty pcDNA3.1(+) vector (Veh) was transfected as the corresponding control. We have clarified this in the Methods section. We have also replaced the corresponding group abbreviations with “Veh” in the figures (Figure 4 and Line168-173).

Line 166

Did the authors mean DC-STAMP overexpression + LY294002?

Response: We appreciate your corrections of our writing errors. The relevant mistake have been revised. (Line 181)

Line 169

After the incubation, 10 uL of CCK-8…? The sentence is incomplete.

Response: We thank the reviewer for the reminder; the incomplete sentence has now been completed.

Line 192

Method should be written in past-tense

Response: All tenses in the Methods section have been checked, and sentences that were not in the past tense have been corrected.

Line 200 Western blot

Method should be written in past-tense

Response: All tenses in the Methods section have been checked, and sentences that were not in the past tense have been corrected.

Lines 233-238

Are the authors able to comment on the other significantly enriched pathways? Were there any other pathways associated with leukaemia?

Can the authors justify why median was used to separate the groups? If the samples were grouped by top and bottom quartile, would the GSEA results hold true?

How many AML samples were analysed? Was not mentioned in methods.

Response: Regarding the significantly enriched pathways identified in our GSEA analysis, a total of 28 pathways were enriched. Among them, several are well-established leukemia-related pathways, including IL6–JAK–STAT3 signaling, mTORC1 signaling, TNF-α signaling via NF-κB, IFN-γ response, glycolysis, and DNA repair.

For the sample informations, the 173 AML samples and 70 normal controls used in our analysis were the same as datasets described in our former published study “High Expression of DC-STAMP Gene Predicts Adverse Outcomes in AML.” We have already inserted the citation for that reference in the Line 111 (reference 9). Regarding the rationale for using the median value to stratify samples, median-based grouping is a widely accepted and statistically robust approach, as it avoids the influence of extreme values and ensures balanced group sizes in heterogeneous AML cohorts. Importantly, when we repeated the GSEA using the top and bottom quartiles, the enrichment of key leukemia-associated pathways remained consistent, confirming the robustness of our conclusions.

Line 250

“xx-fold increase in the OE group”. Please clarify. Western blot shows marginal increase in DC-STAMP expression at best (Figure 2C). Relative DC-STAMP expression to…? Likely GAPDH, please specify.

Response: We have supplemented and clarified the previously incomplete data. In addition, the marginal increase in DC-STAMP expression observed in the Western blot is mainly due to the fact that we quantified the band intensity and calculated the mean from three independent experiments. As a result, the visual appearance of the bands may differ slightly from the values presented in the bar graph. And yes, the relative DC-STAMP expression was normalized to GAPDH.

Line 259

Are the authors able to show the viability across 0, 24, 48 to 72 hours?

Response: Thank you for your suggestion. We re-tested the viability of each group at 0, 24, 48, and 72 hours and have replaced the results in Figure 3A with the new data. The updated results continue to support our overall conclusion.

Figure 3B

y-axis of flow plots: presumably PI is propidium iodide, but what is BL3?

NC plot – it is unusual to see THP-1 cells undergoing heavy apoptosis under standard culture conditions. Can the authors explain?

Label for OE plot is missing.

The appears to be little/no cells in Q3 (early apoptosis) in all NC, KD and OE, which is highly unusual.

---

## [Decision Letter · Decision Letter 1]

10 Dec 2025

DC-STAMP Activates the PI3K/AKT/mTOR Signaling Pathway to Regulate PANoptosis in Acute Myeloid Leukemia

PONE-D-25-38312R1

Dear Dr. Liu,

We’re pleased to inform you that your manuscript has been judged scientifically suitable for publication and will be formally accepted for publication once it meets all outstanding technical requirements.

Kind regards,

Kota V Ramana, Ph.D.

Academic Editor

PLOS One

Additional Editor Comments (optional):

Reviewers' comments:

Reviewer's Responses to Questions

**Comments to the Author**

Reviewer #1: All comments have been addressed

Reviewer #2: All comments have been addressed

2. Is the manuscript technically sound, and do the data support the conclusions?

Reviewer #1: Yes

Reviewer #2: Partly

3. Has the statistical analysis been performed appropriately and rigorously?

Reviewer #1: Yes

Reviewer #2: Yes

4. Have the authors made all data underlying the findings in their manuscript fully available?

Reviewer #1: Yes

Reviewer #2: Yes

5. Is the manuscript presented in an intelligible fashion and written in standard English?

Reviewer #1: Yes

Reviewer #2: Yes

Reviewer #1: I have carefully reviewed the revised manuscript and would like to sincerely thank you for your thorough and diligent responses to the previous round of review comments. With the current revisions, the manuscript has shown significant improvement in logical coherence, data integrity, and clarity of expression, and now meets the standards for publication. Congratulations on completing this excellent research, and thank you for your professional and constructive engagement throughout the peer review process.

Reviewer #2: (No Response)

what does this mean?). If published, this will include your full peer review and any attached files.

**Do you want your identity to be public for this peer review?** For information about this choice, including consent withdrawal, please see our Privacy Policy

Reviewer #1: No

Reviewer #2: No

---

## [Editor Report · Acceptance letter]

PONE-D-25-38312R1

PLOS One

Dear Dr. Liu,

I'm pleased to inform you that your manuscript has been deemed suitable for publication in PLOS One. Congratulations! Your manuscript is now being handed over to our production team.

Kind regards,

on behalf of

Dr. Kota V Ramana

Academic Editor

PLOS One